

# Comparative transcriptome analysis of miRNA in hydronephrosis male children caused by ureteropelvic junction obstruction with or without renal functional injury

Ge Liu, Xin Liu and Yi Yang

Urology Division, Pediatric Surgery Department, Shengjing Hospital of China Medical University, Shenyang, Liaoning, People's Republic of China

## ABSTRACT

MicroRNAs (miRNAs or miRs) are non-coding RNAs that contribute to pathological processes of various kidney diseases. Renal function injury represents a final common outcome of congenital obstructive nephropathy and has attracted a great deal of attention. However the molecular mechanisms are still not fully established. In this study, we compared transcriptome sequencing data of miRNAs of renal tissues from congenital hydronephrosis children with or without renal functional injury, in order to better understand whether microRNAs could play important roles in renal functional injury after ureteropelvic junction obstruction. A total of 22 microRNAs with significant changes in their expression were identified. Five microRNAs were up-regulated and 17 microRNAs were down-regulated in the renal tissues of the hydronephrosis patients with renal function injury compared with those without renal function injury. MicroRNA target genes were predicted by three major online miRNA target prediction algorithms, and all these mRNAs were used to perform the gene ontology analysis and Kyoto Encyclopedia of Gene and Genomes pathway analysis. Then, twelve candidate human and rat homologous miRNAs were selected for validation using RT-qPCR *in vitro* and *in vivo*; only miR-187-3p had a trend identical to that detected by the sequencing results among the human tissues, *in vivo* and *in vitro* experimental models. In addition, we found that the change of miR-187-3p *in vivo* was consistent with results *in vitro* models and showed a decrease trend in time dependence. These results provided a detailed catalog of candidate miRNAs to investigate their regulatory role in renal injury of congenital hydronephrosis, indicating that they may serve as candidate biomarkers or therapeutic targets in the future.

## INTRODUCTION

Ureteropelvic junction obstruction (UPJO) is one of common clinical symptoms causing to persistent hydronephrosis in congenital hydronephrosis children. As a common pediatric urinary malformation, UPJO accounts for the majority of congenital urinary

Corresponding author
Yi Yang, yangy2@sj-hospital.org

tract obstruction in the fetus, infants, and children (*Kohaut & Tejani, 1996*). If there is no effective treatment, UPJO could cause to the progressive renal dysfunction (*Jiang et al., 2017*). UPJO is a congenital or acquired disease. Especially, congenital UPJO leads to chronic kidney disease (CKD) and end-stage renal disease (ESRD) in children and infants (*Chevalier, 2004*; *Harambat et al., 2012*), which represents a progressive irreversible structural damage and/or renal injury. Congenital UPJO affects millions of children's healthy in the world (*Babu, Rathish & Sai, 2015*). Therefore, there is an urgent need to investigate the mechanisms of congenital UPJO onset and progression.

Over the last decade, many studies have been devoted to exploring the molecular mechanisms involved in the progression of CKD. However, the mechanisms of the development of renal fibrosis and functional damage are not clearly understood. MicroRNAs are a class of endogenous, small, non-coding RNAs that play an important role in regulating gene expression at the post-transcriptional level (*Agrawal et al., 2003*; *Krol, Loedige & Filipowicz, 2010*). Previous clinical and experimental animal studies demonstrate that miRNAs play essential roles in the pathogenesis of various renal diseases (*Oba et al., 2010*; *Denby et al., 2014*; *Chung & Lan, 2015*; *Loboda et al., 2016*). However, these studies were either based on animal models or limited to a single miRNA of interest, and these studies did not compare the miRs in animal with that in human.

In this study, we performed the transcriptome sequencing of miRNAs in kidney tissue, and compared the data between normal differential renal function (DRF) and declined DRF infants with ureteropelvic junction obstruction to better investigate the molecular mechanism of renal injury in congenital hydronephrosis.

## MATERIALS & METHODS

### Patients and samples

From August 2016 to June 2017, 12 patients that are younger than 6 months with congenital unilateral hydronephrosis were selected as the research subjects. To confirm the diagnosis of hydronephrosis, all patients underwent urinary ultrasound and magnetic resonance urography (MRU), as well as Diuretic radionuclides (DRS) preoperatively. We divided the patients into two groups depending on their preoperative differential renal function (DRF), patients with normal DRF level ($>45\%$) were classified as normal DRF group while those with declined DRF level ($<30\%$) were considered as abnormal DRF group. All of them received open pyeloplasty, the ureteropelvic junctional obstruction were confirmed by operation and pathological postoperatively. The kidney parenchyma tissues were collected during surgery, according to the study of *Song et al. (2019)*, and the tissues were frozen in liquid nitrogen and stored at $-80\,°C$ for further study. The study was approved by the Ethics Committee of Shengjing Hospital, China Medical University, Shenyang, China (No. 2012PS81K), and the Ethics Committee waived the need for consent.

### Experimental procedure of unilateral ureteral obstruction

The Unilateral Ureteral Obstruction (UUO) is a common prepared animal model in studying obstructive nephropathy. Two litters of 24 newborn Sprague Dawley (SD) rats (weight, 13–18 g, specific pathogen free) were obtained from Liaoning Changsheng

Biotechnology Co. Ltd. (Benxi, China). Pups were kept with their dams in a laboratory animal facility with a constant temperature of $23 \pm 2\,°C$ and relative humidity of $40 \pm 10\%$ under a regular 12 h light/dark cycle. All animal care and experimental protocols were approved by the Committee on the Ethics of Animal Experiments of Shengjing Hospital, China Medical University, Shenyang, China (No. 2018PS228K).

One litter of pups were randomly divided into UUO-2d sham group and UUO-2d group ($n = 6$, per group), the other litter of pups were randomly divided into UUO-5d sham group, and UUO-5d group ($n = 6$, per group). Animal surgical procedures were performed using sterile technique under isoflurane anaesthesia. The pups in the UUO groups underwent ligation of the left ureter within 48 h after birth. Briefly, the left ureter was visualized by a flank incision, ligated with 8–0 silk below the renal pelvis and cut between the two ligatures. The sham group underwent the same surgery except for left ureter ligation. Postoperatively, the pups were returned to their dams.

The criteria were established for euthanizing animals prior to the planned end of the experiment. The pups were sacrificed with an overdose of pentobarbital sodium at 150 mg/kg intraperitoneally (i.p.) to reduce the pain or suffering; At days 2ed and 5th after surgery, the kidneys were harvested, frozen in liquid nitrogen, and stored at $-80\,°C$ until further processing.

## Cell culture and transform growth factor beta 1 (TGF-β1) treatment

HK-2, a human kidney epithelial cell line, was purchased from the Shanghai Institute for Biological Sciences (SIBS), Chinese Academy of Sciences (Shanghai, China) and cultured for less than six months from the time of resuscitation. HK-2 cells were cultivated in Dulbecco's modified Eagle's medium (DMEM) /F-12 1:1 (Hyclone Laboratories, Inc., Logan, UT, USA) supplemented 10% fetal bovine serum supplemented with 1% penicillin/streptomycin at $37\,°C$, 5% $CO_2$ in a humidified incubator. At ∼50% confluency, cells were treated with recombinant human TGF-β1 (10 ng/ml) (Proteintech Group, Inc.) for 24 h, 48 h, 72 h, 96 h, 120 h and 144 h or the vehicle control in DMEM/F12 1:1 for the indicated time period.

## MiRNA sequence

Briefly, the kidney specimens were collected from hydronephrosis patients, and the RNA sequencing libraries were constructed from the extracted and amplified RNA using the standard Illumina library preparation protocols. miRNA-seq was performed on the Illumina HiSeq 2500 platform using PE150 protocol by Genesky Biotechnologies, Inc. (Shanghai, China). Data were collected as previously described as *Liu et al. (2017)* reported. The tool of ASprofle was performed for extraction and comparison of Alternative splicing (AS) events (*Florea, Song & Salzberg, 2013*).

## Prediction of target genes

The target genes of the differentially expressed miRNAs were screened by the starBase v2.0 (http://starbase.sysu.edu.cn) (*Li et al., 2014*), and gene searched in >2 databases simultaneously, such as TargetScan (http://www.targetscan.org), miRTarBase (http://mirtarbase.mbc.nctu.edu.tw) or miRmap databases (http://mirmap.ezlab.org) were predicted to be target genes.

## Functional and pathway analysis

For functional enrichment pathway analysis, gene ontology (GO) which was subdivided into three groups, including biological processes, cellular components, and molecular functions, was applied to analyze the function of the predicted target genes. Similarly, the Kyoto Encyclopedia of Genes and Genomes (KEGG) was used to find out the significant pathway of the predicted target genes. The GO and KEGG analysis were performed using the OmicShare tools, a free online platform for data analysis (http://www.omicshare.com/tools).

## RNA extraction and validation of potential miRNAs using real-time quantitative polymerase chain reaction (RT-qPCR)

Total RNA was extracted from kidney tissues of patients or rats and cultured cells by RNAiso Plus (TaKaRa, Shiga, Japan) according to the manufacturer's instructions. Total RNA (1 μg) was reverse-transcribed to cDNA by using the PrimeScript RT Reagent Kit (TaKaRa, Shiga, Japan) following the manufacturer's protocol. Then qRT-PCR was performed with the SYBR Premix Ex Taq (TaKaRa, Shiga, Japan) and primers (Ribobio, Guangzhou, China) in a final volume of 20 μl. All PCR reactions were done in triplicates and repeated at least three times in a 7500 Real-time PCR machine (Applied Biosystems, Waltham, MA, USA). The relative miRNA levels were normalized and quantified by U6 expression levels for each sample. Gene expression analysis were performed using the $2^{-\Delta\Delta}$ Cycle thresholds ($\Delta\Delta$Ct) method. All primers were designed and synthesized by RiboBio Co., Ltd. (Guangzhou, China).

## Statistical analysis

SPSS version 13.0 (SPSS Inc. Chicago, IL, USA) and GraphPad Prism version 5.0 (GraphPad, San Diego, CA, USA) were used to analyze the data, which are expressed as the mean ± standard deviation (SD). Differential expression of miRNAs in the kidney tissues of patients with UUO in abnormal renal function group compared to those in normal renal function group was detected using two-sided Student's $t$-test. $P < 0.05$ was considered to be statistically significant.

# RESULTS

## Clinical information

All the patients with congenital hydronephrosis were Society of Fetal Urology (SFU) grade 4. Among the six patients who underwent miRNA-seq, age distributions were 6–14 weeks in normal renal function group (median age, 10 weeks) and 7–18 weeks in abnormal renal function group (median age, 10 weeks). Meanwhile, the other six patients who underwent RT-qPCR validation assays, age distributions were 12–20 weeks in normal renal function group (median age, 13 weeks) and 6–22 weeks in abnormal renal function group (median age, 12 weeks). All the patients are male.

## Differentially expressed renal miRNAs in hydronephrosis children with or without renal dysfunction

After comparative analysis of the miRNA-seq data, 22 miRNAs were differentially expressed in kidney tissues of abnormal renal function group compared with normal renal function

**Table 1   Fold change expression of 22 differentially expressed miRNAs in renal tissues of congenital hydronephrosis infants from RNA-seq data.**

| miRNA | DNF>45% | DNF<35% | *p*value | log2FoldChange |
|---|---|---|---|---|
| hsa-miR-7704 | 18.35 ± 16.43 | 2.46 ± 2.39 | 0.003613634 | −2.974923849 |
| hsa-miR-3195 | 33.83 ± 28.28 | 5.03 ± 2.07 | 0.00091878 | −2.753388116 |
| hsa-miR-378a-5p | 226.17 ± 151.78 | 43.13 ± 20.66 | 0.000191141 | −2.377348869 |
| hsa-miR-138-5p | 243.66 ± 165.81 | 47.64 ± 32.7 | 0.001229062 | −2.341418947 |
| hsa-miR-378d | 1232.59 ± 906.44 | 243.52 ± 91.08 | 0.000365338 | −2.337746685 |
| hsa-miR-7641 | 631.08 ± 742.43 | 130.48 ± 28.85 | 0.002161003 | −2.269745727 |
| hsa-miR-3656 | 193.71 ± 183.34 | 43.37 ± 17 | 0.001779968 | −2.153430132 |
| hsa-miR-378c | 2191.15 ± 1535.8 | 524.84 ± 167.88 | 0.000907617 | −2.060900181 |
| hsa-miR-378i | 154.42 ± 84.7 | 40.58 ± 22.45 | 0.003083273 | −1.920454026 |
| hsa-miR-378f | 51.55 ± 29.27 | 13.67 ± 4.74 | 0.004442206 | −1.884860173 |
| hsa-miR-204-3p | 128.99 ± 86.44 | 37.53 ± 12.74 | 0.006674413 | −1.768711872 |
| hsa-miR-187-3p | 375.69 ± 245.62 | 111.26 ± 45.93 | 0.006219278 | −1.752649274 |
| hsa-miR-873-3p | 90.82 ± 57.77 | 26.6 ± 12.54 | 0.008710633 | −1.750311834 |
| hsa-miR-378a-3p | 11471.13 ± 6230.49 | 3543.99 ± 1894.21 | 0.004923984 | −1.694418411 |
| hsa-miR-874-3p | 21027.61 ± 10491.11 | 6795.22 ± 2486.92 | 0.002100211 | −1.629586819 |
| hsa-miR-338-3p | 342.78 ± 215.06 | 113.12 ± 38.23 | 0.009335378 | −1.594507199 |
| hsa-miR-4532 | 1383.65 ± 830.96 | 465.35 ± 202.48 | 0.006808167 | −1.572280994 |
| hsa-miR-212-5p | 66.42 ± 53.3 | 220.92 ± 108.49 | 0.007819275 | 1.718296369 |
| hsa-miR-224-5p | 59.88 ± 53 | 235.67 ± 71.48 | 0.002229173 | 1.955015383 |
| hsa-miR-142-5p | 263.33 ± 188.08 | 1076.71 ± 772.04 | 0.002629789 | 2.027340006 |
| hsa-miR-31-5p | 53.08 ± 56.68 | 326.16 ± 350.43 | 0.002166778 | 2.606070929 |
| hsa-miR-21-5p | 27210.04 ± 18167.04 | 181317.86 ± 212736.06 | 0.001015935 | 2.736284032 |

**Notes.**
Fold change expression of 22 differentially expressed miRNAs (fold change >1.5; $P < 0.01$) in renal tissues of congenital hydronephrosis infants from RNA-seq data.

group (fold change >1.5; $P < 0.01$; Table 1). Among them, five miRNAs were up regulation and 17 miRNAs were down regulation in hydronephrosis children with renal dysfunction, Hierarchical clustered heatmap and Volcano plot showed the expression changes of miRNA between the two groups (Fig. 1).

**Construction of target prediction network of miRNA-mRNA**

To identify the function of the differentially expressed miRNAs, the prediction of miRNA target genes was performed by the starBase v2.0 (http://starbase.sysu.edu.cn) (*Li et al., 2014*). Subsequently, select genes that overlapped at least two of the three major online miRNA target prediction algorithms: TargetScan, miRTarBase and miRmap databases. We performed the 22 differentially expressed miRNAs (Table 1) to predict the targeting genes using TargetScan, miRTarBase and miRmap databases, 1,499 genes (Supplement 1) were obtained with overlapping at least two of the three major online miRNA target prediction websides. All of these predicted 1,499 mRNAs were used to perform the GO and KEGG pathway analysis. As shown in Fig. 2A, the most significant top 5 GO terms in biological process were respectively cellular process, biological regulation, metabolic process, regulation of biological process and response to stimulus. In molecular function,

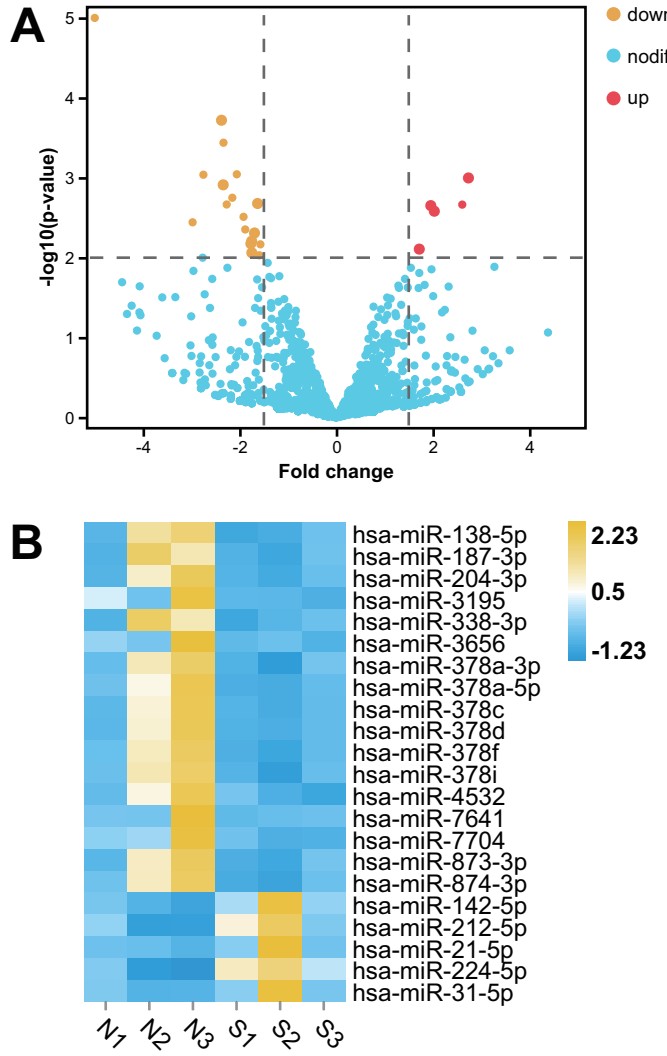

**Figure 1** **Comparative transcriptome analysis of miRNA in human kidney tissues with or without renal functional injury.** Comparative transcriptome analysis of miRNA in human kidney tissues with or without renal functional injury. (A) Volcano plot showed differential expression of miRNA in human kidney tissues between normal and abnormal renal function group. Red dots = Up regulation; Yellow dots = Down regulation. (B) Hierarchical clustered heat map showed the expression patterns of significantly (fold change >1.5 $P < 0.01$) and differentially expressed miRNA: Blue = Down regulation; Yellow = Up regulation; White = No significant difference. Normal renal function group: N1, N2, N3; Abnormal renal function group: S1, S2, S3.

the top 5 GO terms were binding, catalytic activity, transcription regular activity, molecular function regulator, and transporter activity. In cellular component, cell, cell part, organelle, organelle part and membrane were the top five GO terms. A total of 64 distinct pathways with enrichment test $P < 0.05$ were identified according to the KEGG pathway database. In addition, KEGG pathway analysis revealed that the identified miRNAs regulated the signaling pathways of cancer, Cushing Syndrome, Longevity regulation, regulation of pluripotency of stem cells, axon guidance, etc (Fig. 2B).

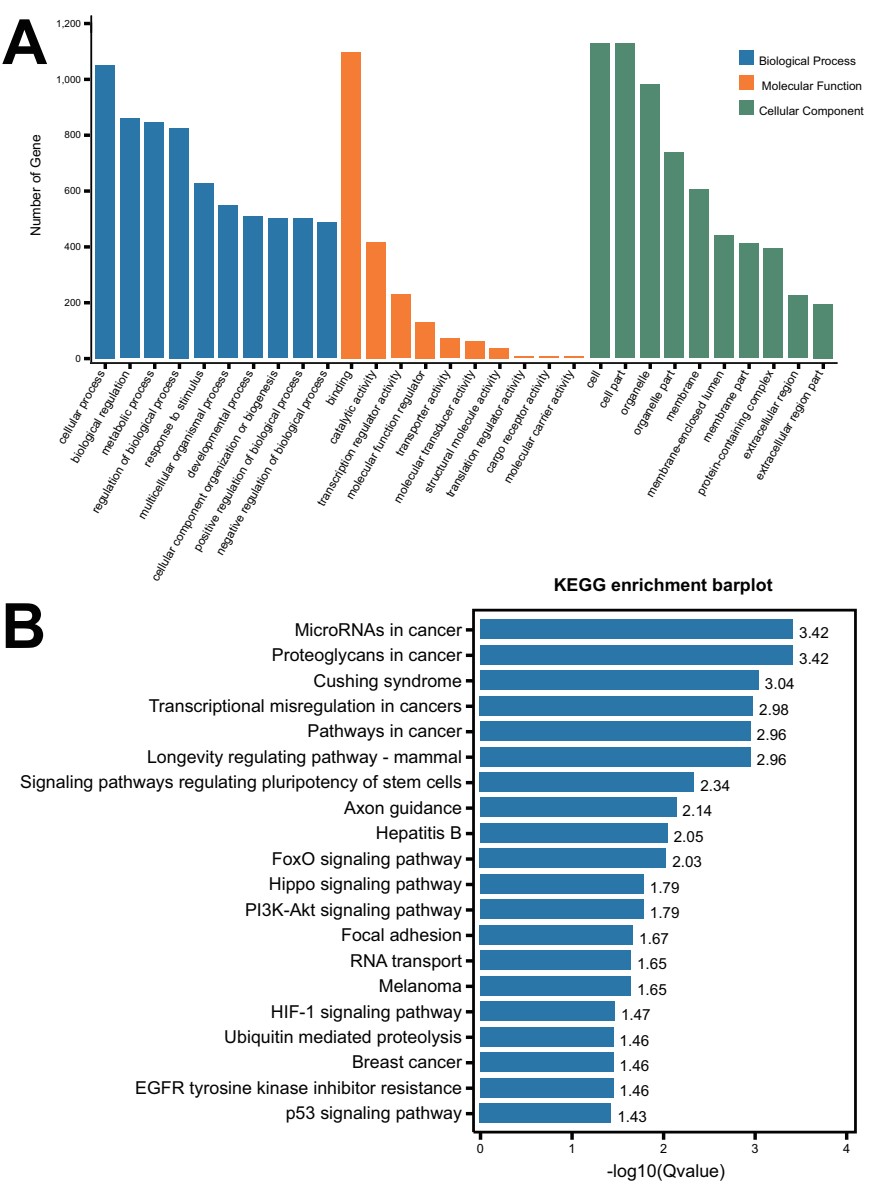

**Figure 2  GO analysis and KEGG pathway analysis of the miRNA-mRNA Network.** (A) GO enrichment analysis of biological process, molecular function and cellular compartments. Blue bars represent the number of biological processes related genes; Orange bars represent the number of molecular function related genes; Green bars represent the number of cellular component related genes; (B) Barplot of KEGG pathway enrichment.

## Validation of the microarray data through RT-qPCR in neonatal rat kidney tissues with UUO

To validate the microarray data, twelve candidate human and rat homologous miRNAs were selected for miRNA expression validation using RT-qPCR, including four up-regulated miRNAs (miR-21-5p, miR-224-5p, miR-142-5p, miR-212-5p) and eight down-regulated

**Table 2  Human and rat homologous miRNAs sequence.**

| miRNA | Human | Rat |
|-------|-------|-----|
| miR-378a-5p | CUCCUGACUCCAGGUCCUGUGU | CUCCUGACUCCAGGUCCUGUGU |
| miR-138-5p | AGCUGGUGUUGUGAAUCAGGCCG | AGCUGGUGUUGUGAAUCAGGCCG |
| miR-874-3p | CUGCCCUGGCCCGAGGGACCGA | CUGCCCUGGCCCGAGGGACCGA |
| miR-378a-3p | ACUGGACUUGGAGUCAGAAGGC | ACUGGACUUGGAGUCAGAAGG |
| miR-187-3p | UCGUGUCUUGUGUUGCAGCCGG | UCGUGUCUUGUGUUGCAGCCGG |
| miR-204-3p | GCUGGGAAGGCAAAGGGACGU | GCUGGGAAGGCAAAGGGACGUU |
| miR-873-3p | GGAGACUGAUGAGUUCCCGGGA | GAGACUGACAAGUUCCCGGGA |
| miR-338-3p | UCCAGCAUCAGUGAUUUUGUUG | UCCAGCAUCAGUGAUUUUGUUGA |
| miR-21-5p* | UAGCUUAUCAGACUGAUGUUGA | UAGCUUAUCAGACUGAUGUUGA |
| miR-224-5p* | UCAAGUCACUAGUGGUUCCGUUUAG | CAAGUCACUAGUGGUUCCGUUU |
| miR-142-5p* | CAUAAAGUAGAAAGCACUACU | CAUAAAGUAGAAAGCACUACU |
| miR-212-5p* | ACCUUGGCUCUAGACUGCUUACU | ACCUUGGCUCUAGACUGCUUACUG |

**Notes.**
*Up-regulated miRNA in miRNA-Seq data.

miRNAs (miR-378a-5p, miR-138-5p, miR-874-3p and miR-378a-3p miR-187-3p, miR-204-3p, miR-873-3p, miR-338-3p) (Table 2). In neonatal rat kidney tissues with 2-day UUO, we found that miR-187-3p, miR-873-3p, and miR-21-5p in twelve selected miRNAs were all up-regulated with statistical significance. Only the trend of change in miR-21-5p was consistent with that observed by miRNA-Seq data (Fig. 3A). Furthermore, in neonatal rat kidney tissues with 5-day UUO, we found that the change trends of one (miR-187-3p) from the down-regulated miRNAs and two from the up-regulated ones (miR-21-5p, miR-142-5p) revealed the same expression tendency as miRNA-Seq data and were statistically significant (Fig. 3B). Therefore, miR-21-5p, miR-187-3p, miR-142-5p were selected for further validation in kidney tissues of patients with congenital hydronephrosis.

## Validation of the microarray data through RT-qPCR in kidney tissues of patients with congenital hydronephrosis

We further analyzed the miRNA expression levels of miR-187-3p, miR-21-5p and miR-142-5p in kidney tissues of patients with congenital hydronephrosis using RT-qPCR. The expression levels of miR-187-3p was significantly down-regulated in patients with abnormal DRF compared to normal DRF ones (fold change = 0.277, $p = 0.044$). Meanwhile, the expression level of miR-21-5p was up-regulated but it was not statistically significant (Fig. 3C). Moreover, there is no significant difference in miR-142-5p expression between abnormal DRF group and normal DRF group (Fig. 3C).

## Validation of the microarray data through RT-qPCR in HK-2 cells induced by TGF-β1 *in vitro*

RT-qPCR analysis revealed that miR-187-3p expression was significantly up-regulated in HK-2 cells after induced by TGF- β1 *in vitro*, and the expression level of miR-187-3p decreased with time of induction (Fig. 3D). As shown in Fig. 3D, the expression levels of miR-187-3p in HK-2 cells induced by TGF-β1 *in vitro* were obviously higher than those of control group with ~10.68-fold, ~6.49-fold, ~4.20-fold at 24, 48, 72 h, respectively.

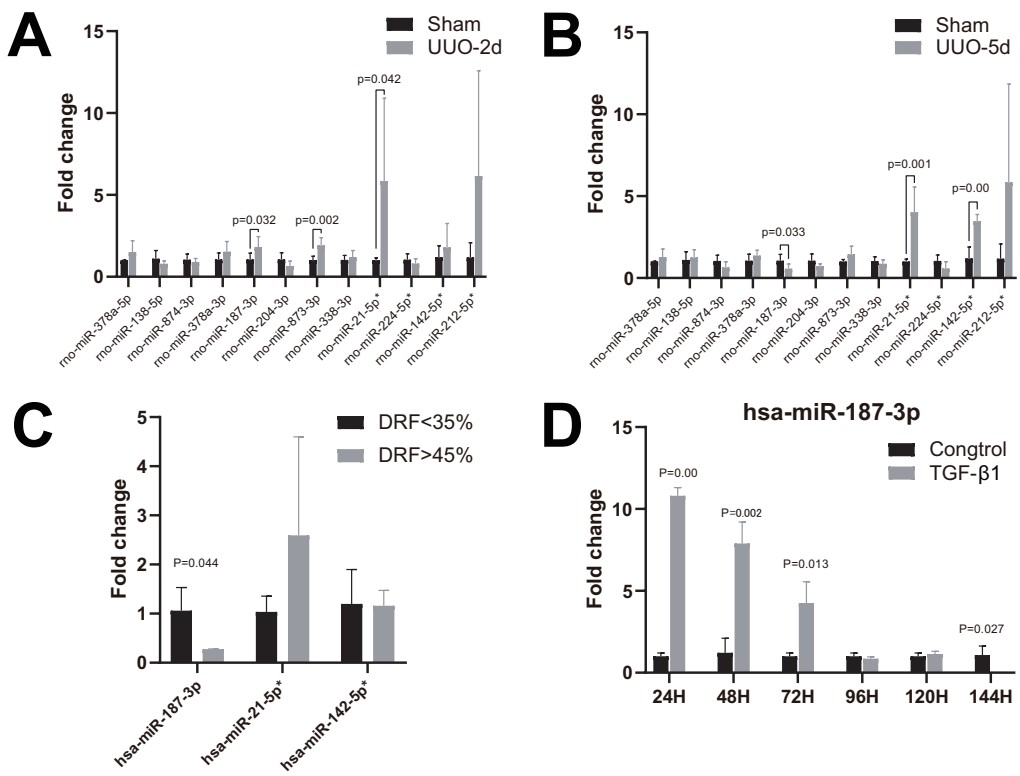

**Figure 3** **RT-qPCR of miRNAs in kidney tissues and HK-2 cell line.** (A) Changes of miRNA expression in kidney tissue on UUO 2d; (B) changes of miRNA expression in kidney tissue on UUO 5d; (C) RT-qPCR of miR-21-5p, miR-142-5p, miR-187-3p in kidney tissues of patients with congenital hydronephrosis; (D) RT-qPCR of miR-187-3p in HK-2 cells induced by TGF-$\beta$1 *in vitro*.

However, at 96 and 120 h under TGF-$\beta$1 treatment, the expression level of miR-187-3p showed no significant difference between the two groups ($p$ >0.05). Induced for 144 h, there was a marked decrease in expression of miR-187-3p (fold change $=$ 0.001; $p=$ 0.027).

## DISCUSSION

Congenital obstructive nephropathy is a principal cause of renal failure in the pediatric population (*Collins et al., 2008*). The pathophysiological mechanism of obstruction nephropathy is very complex and diverse, involving cell apoptosis, oxidative stress, inflammation, and which altogether lead to progressive renal interstitial fibrosis (RIF). The development of RIF is linked to progressive renal injury and CKD (*Falke et al., 2015*). The permanent renal injury, even the obstruction is released by surgical corrections, cannot be prevented in some cases. To date, there are no effective treatments to reverse renal injury after obstruction. Therefore, a better understanding of the pathophysiological mechanism of renal function injury after UUO is required for developing more effective therapeutic strategies.

miRNAs are small non-coding RNA molecules that negatively regulate their target genes at the post-transcriptional level (*Trionfini, Benigni & Remuzzi, 2015*). During the last

decade, a growing number of studies have been undertaken to investigate miRNAs functions (*Trionfini, Benigni & Remuzzi, 2015*). They are involved in numerous physiological and pathological processes, such as organ development, cancer progression and inflammatory diseases. There are many evidences that miRNAs play a crucial part in a variety of kidney diseases, such as renal cancer, IgA nephropathy, acute kidney injury and diabetes nephropathy (*Gottardo et al., 2007*; *Kolfschoten et al., 2009*; *Hennino et al., 2016*; *Qin, Wang & Peng, 2019*).

In humans and animal disease models, altered miRNA expression has been demonstrated in fibrotic disorders of the lung, liver, and kidney (*Vettori, Gay & Distler, 2012*). However, their role in congenital obstructive nephropathy is still in infancy. In order to better understand the mechanism of renal function injury in obstructive nephropathy, we identified the miRNA transcriptome of kidney tissues of congenital hydronephrosis infants with or without renal function injury for the first time in the present study.

Age and complementary food have been shown to influence miRNA expression (*Elhassan, Christie & Duxbury, 2012*; *Meder et al., 2014*; *Ameling et al., 2015*). In order to avoid the influence of different complementary food on the expression of miRNA in kidney, we selected patients younger than 6 months old with hydronephrosis as the research subjects. A total of 22 differentially expressed miRNAs were detected in kidney tissues between the normal and abnormal renal function groups.

The investigations of the miRNA-related functions require further validation in animal models, and miRNA expression profiles may differ in various tissues and species (*Zhou et al., 2018*). Therefore, we selected 12 homologous miRNAs of human and rat for further validation in preparation of kidneys of UUO rat models to confirm the accuracy of miRNA sequencing results.

Because it is difficult to obtain human kidney specimens of obstructive nephropathy, unless the patients have progressed to end-stage renal disease and underwent nephrectomy, the pathological changes within the kidney to the process of renal functional injury cannot be easily addressed. Hence, a variety of experimental models have been developed to study the pathophysiological mechanism in obstructive nephropathy. The UUO model is widely used to study because the procedure has similar characteristics to human obstructive nephropathy (*Martínez-Klimova et al., 2019*). There is inter-individual difference about the obstruction of ureters, therefore, we chose complete unilateral ureteral obstruction in the rat model to induce renal injury. Previously researches have shown that nephrogenesis proceeds postnatally in the neonatal rats or mice (*Hartman, Lai & Patterson, 2007*). This view was supported by *Chevalier, Forbes & Thornhill* in *2009*, who wrote that based on the duration of nephrogenesis, the rat or mouse at birth parallels the midtrimester human, and the 7-day-old rat or mouse is analogous to the full-term human at birth (*Chevalier, Forbes & Thornhill, 2009*). Therefore, the neonatal rat UUO model was chosen for this study. Besides, Chevalier RL's data indicated that relief of the obstruction after two to five days permitted recovery of renal structure and function (*Chevalier et al., 2002*). Therefore, we collected the rat kidney samples on two and five days after UUO, which might be related to renal function injury phase. We performed qRT-PCR to confirm the differential expression of the miRNAs between sham group and UUO groups. The results showed that

the tendencies of miR-874-3p, miR-204-3p, and miR-212-5p aligned with the results in previous miRNA sequence, but these changes did not reach statistical significance. miR-187-3p was down-regulated and miR-21-5p and miR-142-5p were up-regulated, which agreed with the miRNA sequence results. Interestingly, the expression of miR-187-3p was up-regulated in rat kidney tissues on 2 days after UUO and down-regulated on 5 days after UUO.

It has been demonstrated that miR-21 including miR-21-5p plays an important role in renal fibrosis (*Li et al., 2020*). Also, in the study by *Ma et al. (2016)* miR-142-5p targets Smad3 thereby suppressing TGF-ß-induced growth inhibition in cancer cells. The validation results in kidney tissues of patients showed that although the expression of miR-21-5p and miR-142-5p differed between sham rat and UUO rat, no differences were found in the kidney tissues of infants with congenital hydronephrosis. This could be related to the design of our study, the complete UUO model was chosen for this study, which could lead to reduced renal blood flow, interstitial fibrosis and reduced glomerular filtration rate (*Vaughan Jr et al., 2004*). However, the ureteral obstruction is usually not complete obstruction in children with hydronephrosis, the severity of obstruction and the degree of injury are variable, the number of patients was small and there might have been selection bias. These may be the possible reason for this result. Transforming growth factor-β is the strongest known fiber genic factor and plays an important role in the pathogenesis of renal interstitial fibrosis (*Feger et al., 2015*). Treatment of renal epithelial cells with TGF-β1 is a model commonly used to study renal interstitial fibrosis (*Liu, 2004*). TGF- β is also a key EMT inducer that up-regulates important transcription factors in EMT. In this study, we used TGF-β1 to establish a cell fibrosis model *in vitro*. It was worth noting that up-regulation of miR-187-3p in early stage (24–72 h) after TGF-β stimulation has been observed in HK-2 cells, then gradually decreased, besides, the expression of miR-187-3p was down-regulated on day 6 after TGF-β stimulation. We speculate that the expression of miR-187-3p was initiated by UUO, it could play a positive role in protecting renal function, with the progression of the disease, miR-187-3p was consumed gradually till down-regulated by 144 h, indicating that miR-187-3p may serve a role in the development of renal injury. Its expression could be a biomarker of renal function injury in infants with congenital hydronephrosis, However, this hypothesis needs to be further verified using more *in vivo* and *in vitro* experiments.

Following that, we analyzed the predicted 1499 genes from 22 differentially expressed miRNA by GO and KEGG. We demonstrated that some important predicted genes were repeated in top 20 pathways, including PIK3R1, MAPK1, IGF1, IGF1R and AKT3. And their related signaling pathways including FoxO signaling pathway, PI3K-Akt signaling pathway and Hippo signaling pathway, have been reported by other researchers to regulate the renal inflammation and fibrosis in CKD (*Yoon et al., 2014*; *Patel et al., 2019*). Among those differentially expressed miRNAs, miR-187-3p was reported to participate in regulating the many different biological processes, especially in cancer progression (*Dou et al., 2016*; *Sun et al., 2016*; *Zhang et al., 2016*). Moreover, *Zhang et al. (2016)* demonstrated that miR-187 inhibited the epithelial-mesenchymal transition (EMT) process via repressing the TGF-β/Smad pathway through targeting NT5E and PTK6 in colorectal carcinoma;

*Dou et al. (2016)* also demonstrated that miR-187-3p could suppressed the metastasis and EMT of hepatocellular carcinoma. Although the mechanism by which congenital obstructive nephropathy produced renal injury was not completely understood, it has been suggested that TGF-β1-induced epithelial-to-mesenchymal transition plays a key role in renal injury in obstructive nephropathy (*Chevalier et al., 2010*). Epithelial-to-mesenchymal transition was reported to be an important phenotype during CKD. EMT in tubule is a key hallmark in obstructed kidneys. Tubular cells dedifferentiate to fibroblasts and move to the interstitium leading to obstructed kidneys (*Yang & Liu, 2001*; *Iwano et al., 2002*). In the present study, the expression of miR-187-3p was up-regulated in rat kidney tissues after UUO and HK-2 cells after TGF-β stimulation at the early time points, however, miR-187-3p was down-regulated at the late time points. Consistently, we observed that miR-187-3p was down-regulated in renal tissues of congenital hydronephrosis patients with renal function injury. Therefore, miR-187-3p may perform the inhibitory role of EMT to regulate processes of renal injury following ureteral obstruction, and it may be a novel therapeutic target for UUO induced renal injury. However, there is still more validation experiments to investigate the mechanism of miR-187-3p on UJPO.

## CONCLUSIONS

In this study, we have performed a miRNAs transcriptional analysis of kidney tissues of congenital hydronephrosis infants with or without renal function injury, in order to track changes in miRNAs expression that may be associated with renal injury in obstructive nephropathy. These miRNAs could potentially serve as novel biological markers for indicating damaged renal function. In the future, the role of these miRNAs in processes such as apoptosis, proliferation, and differentiation should be determined, as well as its involvement in congenital obstructive nephropathy, and we shall explore the in-depth mechanism to provide additional targets for therapeutic intervention.

## ACKNOWLEDGEMENTS

The authors thank Fujiang Ma and Xueyan Li for their great help in RNA collection and RT-qPCR analysis.

### Funding

This work was supported by the National Natural Science Foundation of China, No. 81571514 and the Key Research and Development Program of Liaoning Province of China, No. 2020JH 2/10300145. The funders had no role in study design, data collection and analysis, decision to publish, or preparation of the manuscript.

### Grant Disclosures

The following grant information was disclosed by the authors:
National Natural Science Foundation of China: 81571514.

Key Research and Development Program of Liaoning Province of China: 2020JH 2/10300145.

## Competing Interests

The authors declare there are no competing interests.

## Author Contributions

- Ge Liu performed the experiments, analyzed the data, prepared figures and/or tables, authored or reviewed drafts of the paper, and approved the final draft.
- Xin Liu analyzed the data, prepared figures and/or tables, authored or reviewed drafts of the paper, and approved the final draft.
- Yi Yang conceived and designed the experiments, analyzed the data, authored or reviewed drafts of the paper, and approved the final draft.

## Human Ethics

The following information was supplied relating to ethical approvals (i.e., approving body and any reference numbers):

The Ethics Committee of Shengjing Hospital, China Medical University, Shenyang, China approved the study (No. 2012PS81K).

## Animal Ethics

The following information was supplied relating to ethical approvals (i.e., approving body and any reference numbers):

The Committee on the Ethics of Animal Experiments of Shengjing Hospital, China Medical University, Shenyang, China approved the study (No. 2018PS228K).

## Microarray Data Deposition

The following information was supplied regarding the deposition of microarray data:

The raw sequence reads are available at NCBI: PRJNA591062.

## Data Availability

The raw measurements of RT-qPCR are available in the Supplementary Files.

## Supplemental Information

Supplemental information for this article can be found online at http://dx.doi.org/10.7717/peerj.12962#supplemental-information.

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
