# Peer review of "Comparative transcriptome analysis of miRNA in hydronephrosis male children caused by ureteropelvic junction obstruction with or without renal functional injury"

_PeerJ, doi:10.7717/peerj.12962_

## Round 0.1 · original submission · Major Revisions

Please follow both reviewers' instructions.

Reviewer 1 ·

Basic reporting

This is a clearly written manuscript that focuses on the molecular biology of pediatric obstructive uropathy which is a relatively common condition that impacts approximately 1:500-1,500 live births. Understanding the spectrum of underlying biomarkers (in the case of the present paper it is miRNAs) may have prognostic significance in determining which patients may progress to chronic kidney disease or end-stage renal disease and which will not.

The construction of the citations in the text and the reference section is confusing. The citations in the text use authors and year. Citations in the reference section are numerical and not in alphabetical order. It would appear that this manuscript has been revised from a version submitted to another journal previously. It is very difficult to ascribe a point made in the text with the appropriate cited literature.

The Figures are fine but the raw data is unreadable. The key animal and IRB compliance documents are in chinese characters and the supplemental data, a critical aspect of the review, is locked and unaccessible.

Experimental design

Well-defined research question and within the scope of the Journal. How to distinguish between children with and without kidney functional deficits as a result of ureteropelvic junction obstruction has significant clinical ramifications.

Methodology is nicely described and the statistical treatment of the data appropriate. The correct databases were utilized for predictive assessments of target genes.

Since the key compliance documents involving approval of animal studies and obtaining human tissue for research are in chinese characters, it is impossible for this reviewer to assess the ethical standards involved.

Validity of the findings

There are several issues outstanding that need to be addressed in a revised manuscript.
The lack of information regarding the miRNA target genes is a significant deficit. There were only 1499 genes that were identified as targets of the resolved miRNAs. They should be listed in a supplemental Excel file. It would be critically important to also indicate if these specific targets are shared in the human vs. rat obstructive diseases.

The paragraph beginning on line 288 is confusing and is of suspect relevance. This section of the discussion mixes KEGG pathway data on tumors with fibrosis and the linkages are not at all clear. The relationship to EMT is appreciated but not well described and while epithelial EMT may (and this is questionable) relate to tubular recovery it does not contribute to interstitial fibrosis which is driven almost exclusively by infiltrating pericytes (papers for Ben Humphreys lab). The purpose of much the the Discussion is muddled and off the issue.

Reviewer 2 ·

Basic reporting

The article titled “Comparative transcriptome analysis of miRNA in hydronephrosis children caused by ureteropelvic junction obstruction with or without renal functional injury” (#57453) by Liu G et al., emphasizes miRNA regulation and its possible role in renal injury in pediatric subjects diagnosed with congenital hydronephrosis.

Overall, the findings were of interest to the peers working in congenital obstructive nephropathy but there were several areas that need authors attention to detail. I have the below comments and concerns regarding the article.

Experimental design

The study design justifies the rationale of this work. As a reviewer from a reader’s point of view the below comments were made to improve the overall reach of this work.

Validity of the findings

The findings are of interest to the specific field

Additional comments

As a reviewer from a reader’s point of view the below comments were made to improve the overall reach of this work.
Major:
1. The introduction is very short, and it is not adequately providing the background of the disease and the significance of the work. The authors may think of providing more information on several aspects relevant to this work like hydronephrosis affects which age group, gender differences if exist, factors leading to or effecting the disease, other miRNA work on renal diseases and UPJO, statistics/number of people affected worldwide, number leading to CKD and deaths etc.
2. Line 62: What is the rationale behind choosing children younger than 6 months, is there any gender bias exist, how many male and how many female subjects participated in this study and if there is gender bias, rationale for the same need to be provided
3. Line 187: Why miR-212-5P is excluded? Is the data in the figure 3a and 3B not significant?
4. Line 195: What about miR-142-5P, it is not mentioned in the results?
5. In Figure 2 legend, more information about the figure need to be provided
Minor:
1. Line 1: Delete space before “Comparative”
2. Line 2: Delete space before “Children”
3. Line 19: Add space before parentheses
4. Line 30: delete “of”
5. Line 34: Correct to “in vivo” and make it italics
6. Change in vitro and in vivo to italics font throughout the manuscript
7. Change to “up regulation and down regulation” throughout the manuscript to main uniformity
8. Line 44: add space before parentheses
9. Line 46: delete space before and after “/” and change to “renal dysfunction”
10. Line 56: Elaborate “DRF” in the first appearance of the manuscript
11. Line 63: Children below 6 months were addressed as research “OBJECTS” which is ethically not correct. Change to “subjects”
12. Line 63: Correct to “diagnosis”
13. Line 70: Is there a procedural name to collect the kidney parenchyma? If so, mention
14. Line 71: delete comma
15. Line 72: Delete space in the parentheses (before No. and after K)
16. Line 74: Elaborate UUO at its first appearance in the text
17. Line 75: Add space after “18”
18. Line 84: Correct “Anesthesia”
19. Line 90: Change to “Frozen in liquid nitrogen”
20. Line 91-93: The sentence does not read well. Try to amend this by “in order to reduce the pain or suffering”
21. Line 102 and 103: Add space between numbers and Hours. For e.g.: 24 h
22. Line 107: Correct to amplified
23. Lines 116, 117 and 118: delete space before and after the website links that are in the parentheses
24. Line 121: delete space before and after “GO”
25. Line 125: Change was to were
26. Line 130: Correct to instructions and add space between 1 and µg
27. Line 134: Correct 20 ul to 20 µl
28. Line 144: Correct to significant
29. Line 159: Correct to Hierarchical
30. Line 158-159: 5 miRNA up regulated and 17 miRNA down regulated but in which group? And list those miRNAs in the results
31. Line 163: delete space before and after the website link that is in the parentheses
32. Line 166-173: Discuss the different N and S groups genes that are differentially regulated and name those miRNAs from the data here in the results
33. Line 179: name the 4 and 8 differentially regulated miRNAs from the data here in the results
34. Line 180-181: Move the 3 miRNAs names in parentheses next to the number 3 in line 180
35. Line 186: Correct to Statistically
36. Line 191: Change to “analyzed”
37. Line 191: change to “levels of”
38. Line 191: Add commas next to different miRNAs and remove the special character that is inserted instead of comma. Add “and” next to miR-21-5p
39. Line 195: Delete extra space in the parentheses
40. Line 199 and 201: in vitro to italics font
41. Line 202 and 204: add space between 72 and hours and 144 and hours
42. Line 208: Correct to Obstructive
43. Line 221: Correct to “Progression” and add “and” before inflammatory diseases
44. Line 226: Add space before However
45. Line 234: Correct “Object” to “subjects”
46. Line 236: Correct to “groups”
47. Line 238: Add space after species
48. Line 248: Add “the” before rat
49. Line 249: Change to “Previously”
50. Line 250: Chevalier RL et al., which year? Mention it here
51. Line 253: Add space before parentheses
52. Line 254: Add et al., after Chevalier RL
53. Line 263: Delete comma after UUO
54. Line 276-277: Add space before parentheses; before “Treatment” and add space after fibrosis
55. Line 278: correct to TGF-β1 and in vitro in italics
56. Line 279: Add space after 72 hours and correct TGF-β
57. Line 283: correct to 144 h
58. Line 286-287: Italics to in vitro and in vivo
59. Line 291: Correct to Functional
60. Line 302: Correct to Researchers and change have to has
61. Line 310: Add space before parentheses
62. Line 313: Correct to “it”
63. Table 1: Line 3: Delete space before “fold” and after 0.01 in the parentheses
64. Table 2: Line 1: Correct to “Human”
65. Table 2: Why few residues were colored in “red”? this needs to be mentioned in the legend
66. Figure 1 Legend: Correct to “Hierarchical”; add space in between heatmap; delete space before and after fold change values in the parentheses
67. Figure 1 Legend: Add “and” before differentially
68. Figure 1 Legend: Change up-expression and under-expression to Up and down regulation, consistent with the text.
69. Figure 1 Legend: (A) the “nodiff” color in the figure is not mentioned in the legend
70. Figure 2: legend need more details. There is no adequate information about the figure
71. Figure 3 legend: Delete space after “UUO 2d” and “;”
72. Figure 3 legend: invitro in italics
73. Figure 3D: Correct “Control” spelling in the figure

---

## Round 0.2 · accepted · Accept

I am writing to inform you that your manuscript - Comparative transcriptome analysis of miRNA in hydronephrosis male children caused by ureteropelvic junction obstruction with or without renal functional injury - has been Accepted for publication. Congratulations!

Reviewer 1 ·

Basic reporting

This manuscript has been adequately revised in accord with my previous suggestions. Acceptance is recommended.

Experimental design

This manuscript has been adequately revised in accord with my previous suggestions. Acceptance is recommended.

Validity of the findings

This manuscript has been adequately revised in accord with my previous suggestions. Acceptance is recommended.

Additional comments

This manuscript has been adequately revised in accord with my previous suggestions. Acceptance is recommended.